# Realization of monolayer ZrTe₅ topological insulators with wide band gaps

Yong-Jie Xu[1,7], Guohua Cao[2,7], Qi-Yuan Li[1], Cheng-Long Xue[1], Wei-Min Zhao[1], Qi-Wei Wang[1], Li-Guo Dou[1], Xuan Du[1], Yu-Xin Meng[1], Yuan-Kun Wang[1], Yu-Hang Gao[1], Zhen-Yu Jia[1], Wei Li[3], Lianlian Ji[3], Fang-Sen Li [3], Zhenyu Zhang [2,4], Ping Cui [2,4] ✉, Dingyu Xing[1,5] & Shao-Chun Li [1,4,5,6] ✉

Two-dimensional topological insulators hosting the quantum spin Hall effect have application potential in dissipationless electronics. To observe the quantum spin Hall effect at elevated temperatures, a wide band gap is indispensable to efficiently suppress bulk conduction. Yet, most candidate materials exhibit narrow or even negative band gaps. Here, via elegant control of van der Waals epitaxy, we have successfully grown monolayer ZrTe₅ on a bilayer graphene/SiC substrate. The epitaxial ZrTe₅ monolayer crystalizes in two allotrope isomers with different intralayer alignments of ZrTe₃ prisms. Our scanning tunneling microscopy/spectroscopy characterization unveils an intrinsic full band gap as large as 254 meV and one-dimensional edge states localized along the periphery of the ZrTe₅ monolayer. First-principles calculations further confirm that the large band gap originates from strong spin−orbit coupling, and the edge states are topologically nontrivial. These findings thus provide a highly desirable material platform for the exploration of the high-temperature quantum spin Hall effect.

Topological insulators (TIs) hold an insulating band gap in the bulk and a time-reversal symmetry-protected gapless state on the boundary[1–3]. The two-dimensional (2D) version of a TI, namely, 2D TI, features the one-dimensional (1D) helical edge states as the conductive channels. Due to spin−momentum locking, the backscattering on nonmagnetic impurities is strictly prohibited in the 1D edge states, thus leading to a dissipationless spin current, as characterized by the quantized spin Hall conductance[4,5]. The quantum spin Hall (QSH) effect was first predicted in graphene if the inclusion of spin−orbit coupling (SOC) opens a band gap at the Dirac cone[6]. It was then experimentally realized in HgTe[7,8] and InAs/GaSb[9] quantum wells, but the observation of the QSH effect demands cryogenic temperatures to suppress thermal excitation and bulk conduction, due to the narrow band gaps of ~10 meV. To realize the high-temperature QSH effect (up to room

temperature), substantial efforts have been made to search for alternative QSH materials with wide band gaps[10–27]. However, most of the discovered QSH monolayers thus far exhibit either narrow or even negative SOC gaps[22,23,28–32].

The weak three-dimensional (3D) TIs can be regarded as the stacking of the 2D TI sheets, such as bulk ZrTe₅ and Bi₄X₄ (X = Br, I)[15,21,26,33], while their 3D nature prevents the direct observation of the QSH effect. The topological nature of bulk ZrTe₅ has been characterized by the topological edge states at the side walls (step edges) and a bulk band gap in the terrace of the top surface[21,34]. Later on, the topological nature of bulk ZrTe₅ was also found to be sensitively dependent on the lattice parameters[35], and exotic phenomena have been discovered in bulk ZrTe₅, such as the chiral magnetic effect, anomalous Hall effect, 3D quantum Hall effect, and log-periodic

[1]National Laboratory of Solid State Microstructures, School of Physics, Nanjing University, Nanjing, China. [2]International Center for Quantum Design of Functional Materials (ICQD), University of Science and Technology of China, Hefei, China. [3]Vacuum Interconnected Nanotech Workstation, Suzhou Institute of Nano-Tech and Nano-Bionics, Chinese Academy of Sciences, Suzhou, China. [4]Hefei National Laboratory, Hefei, China. [5]Collaborative Innovation Center of Advanced Microstructures, Nanjing University, Nanjing, China. [6]Jiangsu Provincial Key Laboratory for Nanotechnology, Nanjing University, Nanjing, China. [7]These authors contributed equally: Yong-Jie Xu, Guohua Cao. ✉e-mail: cuipg@ustc.edu.cn; scli@nju.edu.cn

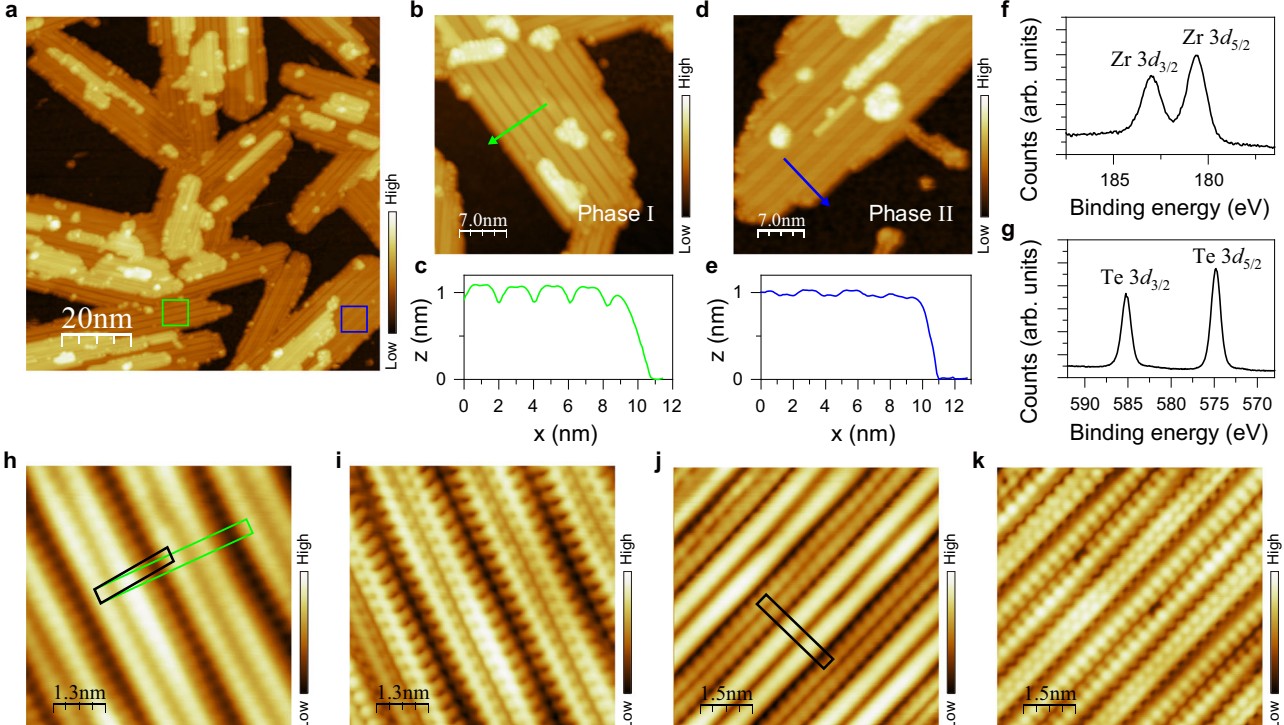

**Fig. 1 | Epitaxial ZrTe₅ monolayers grown on the BLG/SiC(0001) substrate.**
**a** Surface topographic image of the single-layer ZrTe₅ on the BLG/SiC substrate ($100 \times 100$ nm²). Bias voltage $U = +1.0$ V, tunneling current $I_t = 30$ pA. The green and blue squares mark the areas of phases I and II, respectively. **b, d** Zoomed-in topographic images ($35 \times 35$ nm²) of phases I and II, respectively. $U = +1.0$ V, $I_t = 30$ pA. **c, e** Line-scan profiles taken along the arrowed lines in (**b**) and (**d**), illustrating the lattice periods and step heights of phases I and II of the ZrTe₅ monolayers. **f, g** X-ray photoelectron spectroscopy (XPS) results of Zr 3d and Te 3d peaks. **h, i** Atomically resolved topographic images ($6.5 \times 6.5$ nm²) of a phase-I region. **h** $U = +500$ mV, $I_t = 400$ pA, and (**i**) $U = +290$ mV, $I_t = 400$ pA. **j, k** Atomically resolved topographic images ($7.5 \times 7.5$ nm²) of a phase-II region. **j** $U = -300$ mV, $I_t = 500$ pA, and (**k**) $U = +600$ mV, $I_t = 500$ pA. The black parallelogram and green rectangle in (**h**) mark the primitive unit cell and extended orthorhombic cell of phase I, respectively. The black rectangle in (**j**) marks the primitive unit cell of phase II.

oscillations, etc.[36–42]. Even though the ZrTe₅ monolayer has been predicted to be a wide-gap QSH candidate[15], it has not yet been experimentally fabricated to date. In general, van der Waals (vdW) epitaxy is an appealing approach to achieve naturally occurring monolayers, which facilitates the synthesis of quasi-freestanding monolayers and ensures compatibility with device applications[22,23]. However, achieving the vdW epitaxy of ZrTe₅ monolayers is very difficult, as it strongly requires the simultaneous suppression of the formation of other more stable Zr-Te compounds such as ZrTe₂ and ZrTe₃. As reference systems, the epitaxial ZrTe₂ monolayer has been investigated in previous studies[43,44].

In this work, via delicately tuning the epitaxy process, we found the ZrTe₅ monolayer can be only successfully obtained within the rather narrow windows of temperature and flux ratio. Different from its bulk counterpart, the epitaxial ZrTe₅ monolayer exhibits two allotrope isomers with distinct intralayer alignments of trigonal ZrTe₃ prisms. By combining scanning tunneling microscopy/spectroscopy (STM/STS) measurements with first-principles calculations, we determined the atomic structures of the two ZrTe₅ isomers and revealed an intrinsic SOC gap as large as ~ 254 meV in the ZrTe₅ monolayers. We further discovered the 1D gapless edge states localized along the periphery of the ZrTe₅ monolayer, as verified to be topologically nontrivial by our calculations. These findings render the ZrTe₅ monolayer a promising material for demonstrating the high-temperature QSH effect.

## Results
### Epitaxial growth of ZrTe₅ monolayers
To successfully obtain the epitaxial ZrTe₅ monolayer, fine-tuning the epitaxy parameters is found to be crucial. The substrate temperature

for ZrTe₅ growth has to be slightly higher than that for Te crystallization on the surface to avoid the formation of redundant Te islands, and lower than those for ZrTe₃ and ZrTe₂ crystallization to avoid the formation of ZrTe₃ and ZrTe₂ monolayers. Figure 1a shows a typical STM topographic image of the epitaxial ZrTe₅ monolayer on the bilayer graphene (BLG)/SiC(0001) substrate (see Supplementary Figs. 1 and 2 for more details about the epitaxial growth). The epitaxy takes a 2D growth mode, and the surface is dominated by the ZrTe₅ monolayer, as verified by X-ray photoelectron spectroscopy (XPS) measurements shown in Fig. 1f, g. In contrast to bulk ZrTe₅, the ZrTe₅ monolayer is composed of two different structures, as marked by the squares in Fig. 1(a). These two structures are named Phase I [see Fig. 1b] and Phase II [see Fig. 1d] in the following. The measured step heights, as plotted in Fig. 1c, e, are ~ 10.0 Å for both phases, larger than the interlayer lattice constant of ~8.0 Å along the $b$-axis for the bulk (the definition of the axis conforms to the convention for bulk ZrTe₅)[21], indicating a larger vdW distance between the ZrTe₅ monolayer and BLG/SiC substrate. Our large-scale STM data further demonstrate that the orientations of ZrTe₅ monolayers in both phases are randomly distributed, regardless of the BLG/SiC substrate, further indicating a rather weak interlayer vdW interaction (more data can be found in Supplementary Fig. 3). Therefore, these epitaxial ZrTe₅ monolayers are expected to host the quasi-freestanding electronic structures. It is noteworthy that the brighter contrast regions on top of the ZrTe₅ monolayers, as shown in Fig. 1a, are the second layer of ZrTe₅ islands. Besides phases I and II that dominate the second layer of ZrTe₅, the bulk phase of ZrTe₅ also starts to appear in the second layer (more experimental data and first-principles calculations for the bilayers can be found in Supplementary Figs. 4–7). Due to the much smaller sizes and complicated

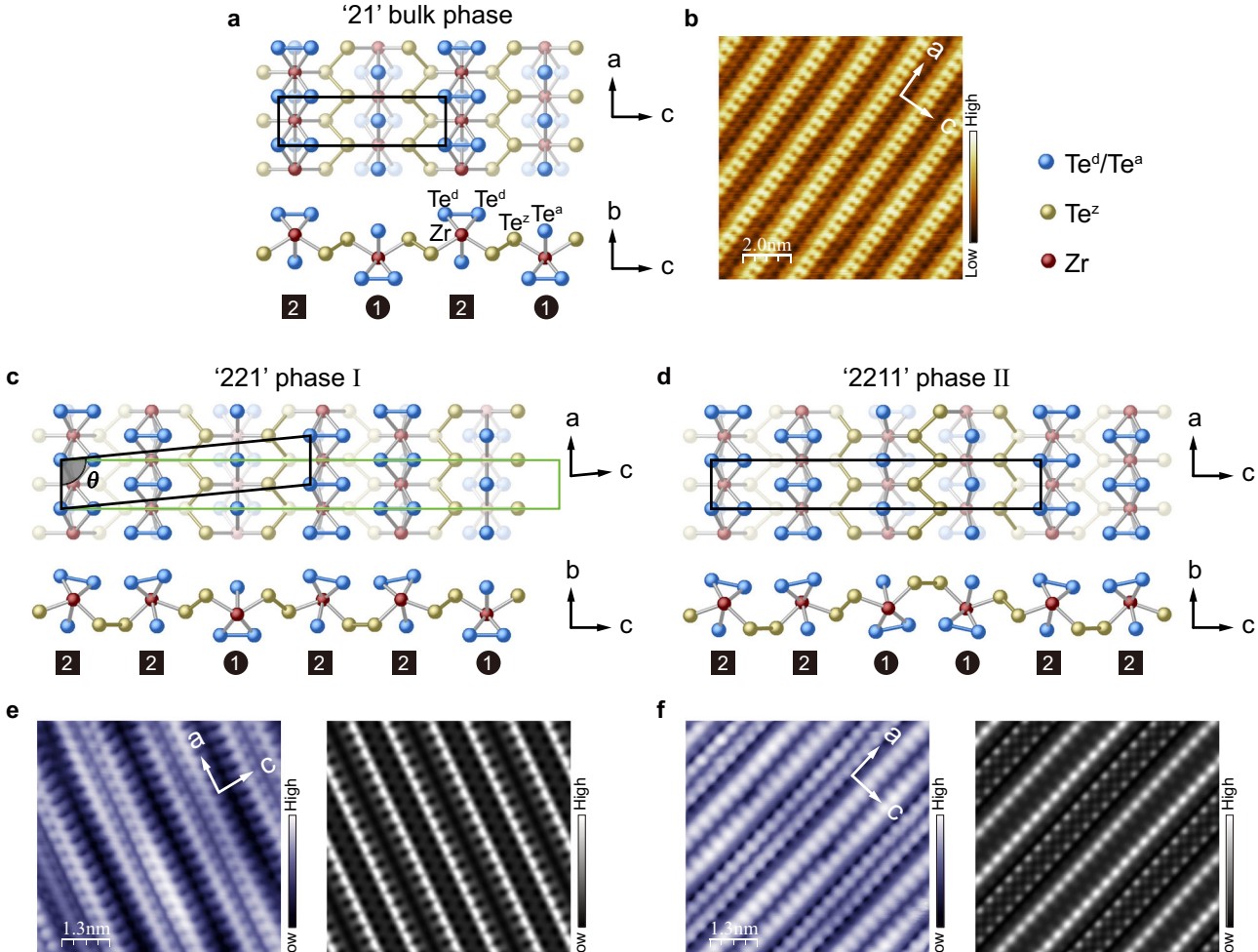

**Fig. 2 | Crystal structures of the ZrTe$_5$ monolayers. a** Top and side views of the crystal structure of bulk ZrTe$_5$, namely, the '21' configuration. The labels '2' indicate the ZrTe$_3$ prims with the Te dimer on top, and labels '1' indicate the ZrTe$_3$ prims with the apical Te atom on top. **b** Atomically resolved topographic image (10 × 10 nm$^2$) taken on the surface of a bulk ZrTe$_5$. $U = +350$ mV, $I_t = 130$ pA. **c, d** Top and side views of the crystal structures of phases I ('221') and II ('2211'), respectively. The black parallelogram in (**c**) and rectangle in (**d**) represent the primitive unit cells of phases I and II, respectively. The green rectangle in (**c**) marks the extended orthorhombic cell that is twice in size of the black parallelogram cell. The angle between the $a$ and $c$ axes in the primitive unit cell is labeled as $\theta$. **e** Comparison between the experimental STM image (left panel, $U = +300$ mV) and simulated image (right panel) for phase I. **f** Same as (**e**) but for phase II. The experimental STM image was obtained at $U = -500$ mV.

interfacial structures, the second layer ZrTe$_5$ is out of the focus of the present work.

The atomically resolved STM images in Fig. 1h–k clearly present the topmost Te atoms of the ZrTe$_5$ monolayer in phases I and II (more data can be found in Supplementary Figs. 8 and 9). Both phases feature bright stripes aligned parallel to the $a$-axis. Different from the bulk ZrTe$_5$ surface that exhibits a single stripe per unit cell[21], the epitaxial ZrTe$_5$ monolayer consists of double stripes per unit cell. The lattice period along the stripes is ~ 3.9 Å for both phases, close to that of bulk ZrTe$_5$ (~ 4.0 Å)[21]. However, the lattice periods along the perpendicular direction are ~ 21.0 and ~ 26.8 Å for phase I and phase II, respectively, ~ 1.5 and 2 times that of bulk ZrTe$_5$ (~ 13.9 Å)[21]. More details can be found in Supplementary Table 1. It is well known that the mismatch of lattice constants and rotational alignments between the epilayer and substrate, e.g., in the presence of a moiré superstructure, can give rise to a difference in the topographic appearances as well. However, the statistics of our STM data show that the morphologies of both phases I and II are independent on their lattice orientations. Moreover, both phases can even coexist in one monolayer island with the same orientation (more details can be found in Supplementary Fig. 3). Thus, it is concluded that phases I and II of ZrTe$_5$ monolayers are intrinsically

distinct lattice structures, rather than different appearances of the same lattice structure due to substrate interactions.

Before further investigating the atomic structures of the epitaxial ZrTe$_5$ monolayers, we give a brief review of the crystal structure of bulk ZrTe$_5$[45]. As shown in Fig. 2a, bulk ZrTe$_5$ crystallizes in an orthorhombic layered structure with the space group of $C_{mcm}$ ($D_{2h}^{17}$)[15]. The basic building block of ZrTe$_5$ is the trigonal prismatic ZrTe$_3$ chain, which is composed of a dimer of Te$^d$ atoms and an apical Te$^a$ atom surrounding a central Zr atom. The ZrTe$_3$ prisms run along the $a$-axis and are interconnected by parallel zigzag Te$^z$ atomic lines to form a 2D sheet in the $a$-$c$ plane, while the $a$-$c$ planes stack along the $b$-axis via vdW forces. In particular, the orientation of the ZrTe$_3$ prisms alternates up and down within the $a$-$c$ plane. Here we designate the prism with Te$^a$ on top as '1' and the inverted prism with a Te$^d$ dimer on top as '2', as illustrated in Fig. 2a. Accordingly, the $a$-$c$ plane of bulk ZrTe$_5$ can be referred to as the '21' configuration.

We now compare the atomic structure of the phase-I monolayer with the $a$-$c$ plane of bulk ZrTe$_5$. For bulk ZrTe$_5$, the bright stripes in the STM images correspond to the top Te$^d$ dimers, namely, '2'-type ZrTe$_3$ prisms, as shown in Fig. 2b[21,46]. Given that the lattice constant of phase I along the $c$ axis is ~ 1.5 times that of the bulk and there exist double

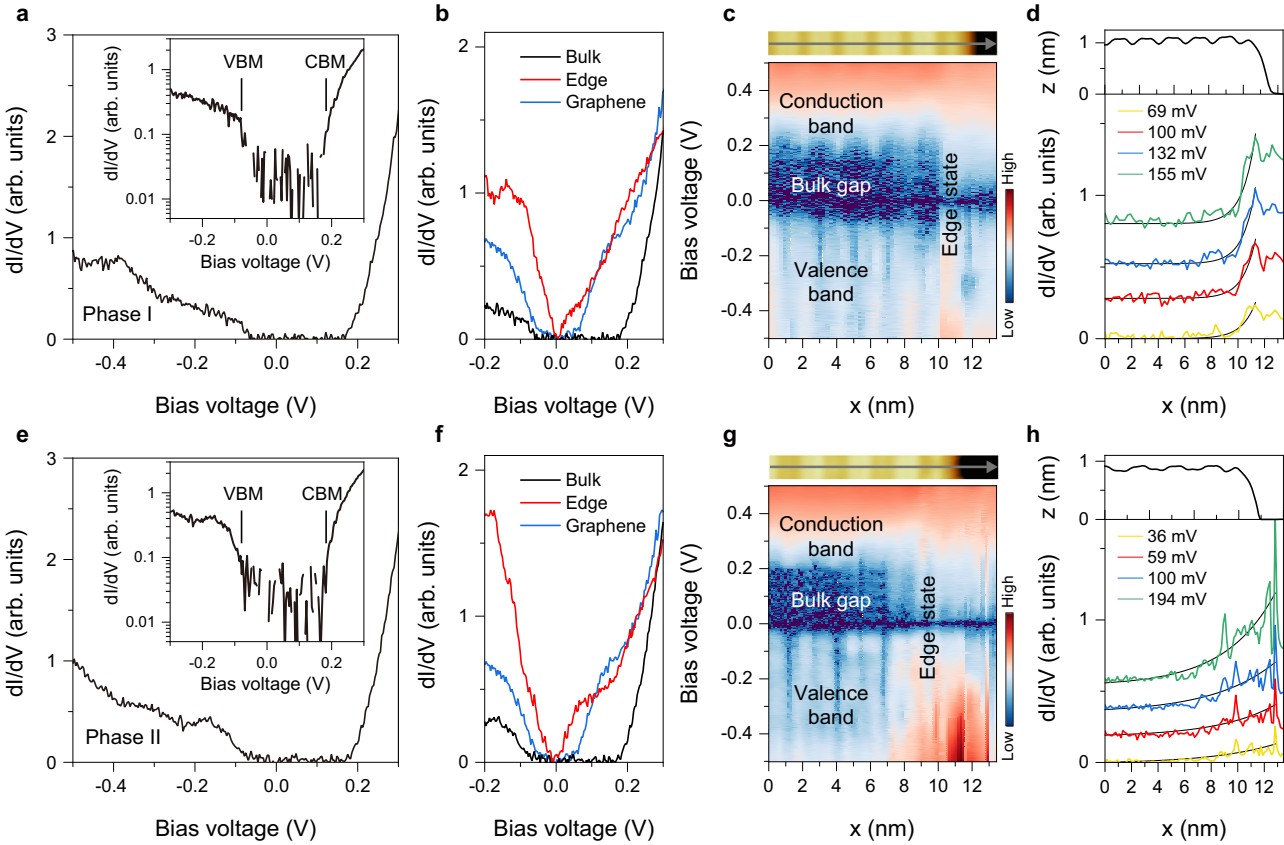

**Fig. 3 | Scanning tunneling spectroscopies on the ZrTe₅ monolayers.**
**a** Differential conductance d*I*/d*V* spectra taken on the phase-I terrace away from the step edge ($U = +500$ mV, $I_t = 100$ pA, the ac modulation voltage $U_{mod} = 8$ mV). Inset: the same spectrum as (**a**) plotted with a logarithmic scale of the *y*-axis and a smaller linear scale of the *x*-axis. The positions of the valence band maximum (VBM) and conduction band minimum (CBM) are marked. **b** Comparison of two representative d*I*/d*V* spectra taken at the phase-I terrace (black) and at the step edge (red). The d*I*/d*V* spectrum taken at the bare graphene substrate (blue) is also plotted for comparison. **c** Spatially resolved d*I*/d*V* spectra (lower panel, $U = +500$ mV, $I_t = 100$ pA, $U_{mod} = 8$ mV) taken across the step edge along the gray arrowed line in the phase-I image (upper panel). **d** Decay of the edge state into the terrace from the step edge in phase I, as manifested by the line-cuts extracted from (**c**) at selective bias voltages. The according topographic line-scan profile is plotted in upper panel for comparison. The black solid lines in the bottom panel show the fit to an exponential decay as a guide to the eye. **e**–**h** Same as (**a**–**d**) but for phase II.

stripes per unit cell, phase I is expected to have two '2'-type prisms and one '1'-type prism per unit cell. Based on our experimental observations and first-principles calculations, the optimized crystal structure of phase I is determined to be the '221' configuration, as shown in Fig. 2c. The calculated lattice parameters are summarized in Supplementary Table 1. The phonon spectrum of the '221' configuration obtained by using the finite displacement method[47] (see Supplementary Fig. 10) shows negligible imaginary frequencies in the whole Brillouin zone, suggesting that the '221' configuration is dynamically stable. The corresponding simulated STM image [Fig. 2e, right panel] is in good agreement with the experimental result [Fig. 2e, left panel]. Specifically, the unit cell of the '221' configuration is a parallelogram containing three ZrTe₃ prims. Alternatively, the parallelogram primitive cell can also be transformed into an extended orthorhombic cell (green rectangle) containing six formula units, as shown in Fig. 2c. Since the Te$^z$ zigzag linkage requires that two neighboring prisms have to be arranged with a half-period shift along the *a*-axis, the angle of the parallelogram ($\theta$) can be described by $\sin(\theta\text{-}90°) = a/2c$ (where *a* and *c* are the lattice constants along the *a* and *c* axes, respectively). The calculated value of $\theta$ is ~95.57°, in good agreement with the experimental measurement (see Supplementary Fig. 11 for more details).

The morphology of phase II is very similar to that of phase I, except for the extra space of an atomic line between two double stripes [see Fig. 1h–k], which may indicate the presence of an extra '1'-type prism between two double '2'-type prisms. Based on our

experimental observations and first-principles calculations, the optimized structure of phase II is determined to be the '2211' configuration, as shown in Fig. 2d, containing two adjacent '2'-type prisms and two adjacent '1'-type prisms per unit cell. The calculated lattice constants are summarized in Supplementary Table 1. Again, negligible imaginary frequencies in the whole phonon spectrum shown in Supplementary Fig. 12 indicate that the '2211' configuration is also dynamically stable. The simulated STM image [Fig. 2(f), right panel] is again in good agreement with the experimental result [Fig. 2(f), left panel].

### Electronic structures of epitaxial ZrTe₅ monolayers

To examine the electronic structures of the ZrTe₅ monolayer, we measured the differential conductance d*I*/d*V* spectra at the atomic scale. The typical d*I*/d*V* spectra, taken randomly on the phase-I terrace, but away from the step edge, are plotted in Fig. 3a (more data can be found in Supplementary Fig. 13). Taking the difference in energy between the intersection points for conduction and valence bands[48], a full gap of as large as ~235 ± 5 meV is identified. To verify the topological nature of the energy gap, we checked the topological bulk-boundary correspondence. Two typical spectra taken on the terrace and the step edge are plotted in Fig. 3b. The spectrum taken on the graphene substrate is also plotted for comparison. The d*I*/d*V* spectrum taken at the step edge shows a gapless 'V'-shaped edge state spanning the whole gap region. Moreover, a dip at the Fermi energy is observed, which may originate from the electron-electron correlation effect,

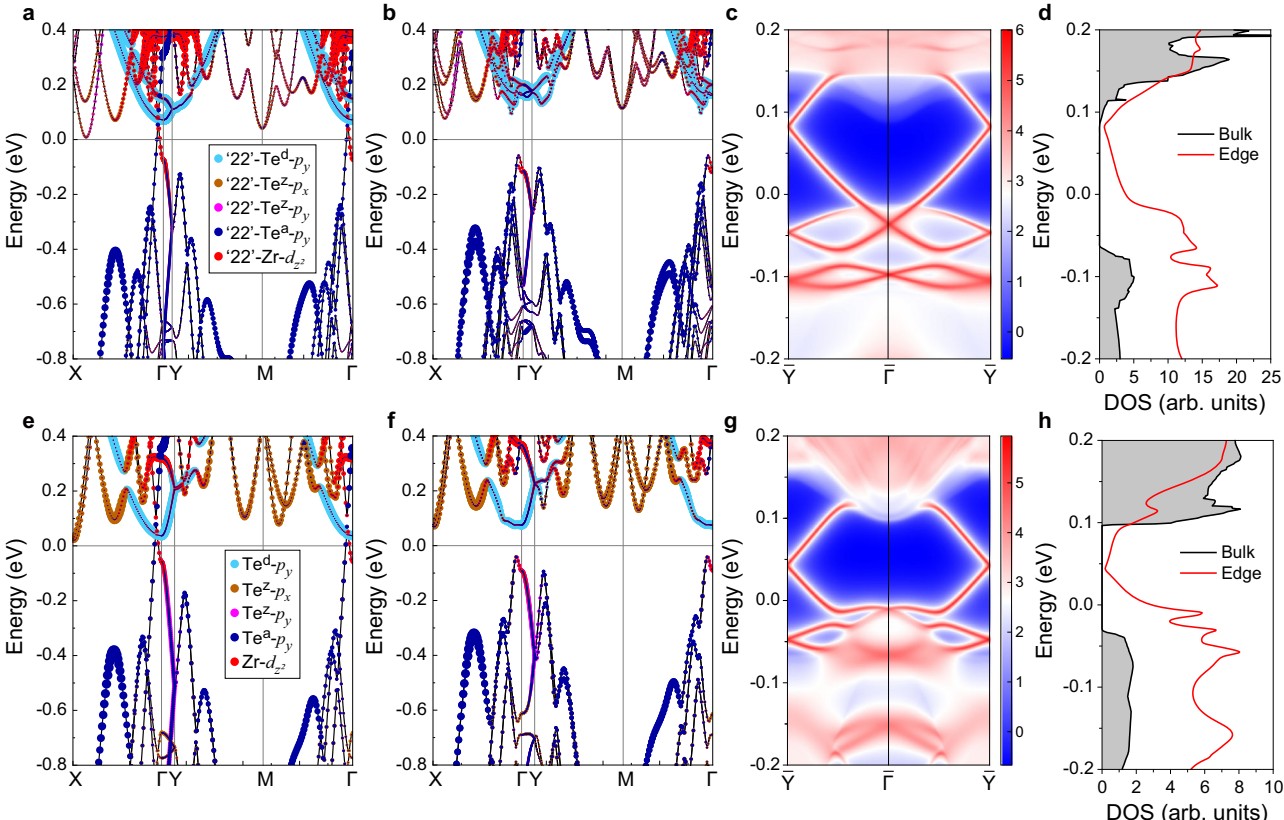

**Fig. 4 | Calculated electronic and topological properties of ZrTe₅ monolayers. a, b** Orbital-decomposed band structures without and with the SOC for phase I. Here, the orthorhombic cell in Fig. 2c is used. The colored balls represent the atomic orbital compositions, and the sizes of the balls are proportional to their contributions, with '22'-Te$^d$-$p_y$, '22'-Te$^z$-$p_x$/'22'-Te$^z$-$p_y$, '22'-Te$^a$-$p_y$, and '22'-Zr-$d_{z^2}$ representing Te$^d$, Te$^z$, Te$^a$ atomic $p$-orbitals, and Zr atomic $d$-orbitals in '22'-type

prisms, respectively. **c** Topological edge states of phase I in a semi-infinite slab perpendicular to the prism stripe (namely, perpendicular to the $a$-axis). The color bar represents the dimensionless magnitude of the projected edge density of states (DOS), with the warmer colors denoting a higher local DOS (LDOS) and the blue region denoting the bulk band gap. **d** Calculated LDOS of the bulk (gray shaded) and edge (red) states of phase I. **e–h** Same as (**a–d**) but for phase II.

as reported in other QSH materials[20,23–25,49,50]. The spatially resolved d$I$/d$V$ spectra taken along a line perpendicular to the step edge, as plotted in Fig. 3c, unambiguously show that the gapped spectral feature evolves sharply into a gapless state, thus indicating that the gapless state is localized right at the periphery of the ZrTe₅ monolayer. The line-cut profiles extracted from Fig. 3c, as plotted in Fig. 3d, show an exponential decay of the edge state into the bulk gap, with a characteristic length of ~1 nm. The spectral results for phase II are phenomenally similar to those for phase I, as shown in Fig. 3e−h, and will not be redundantly described in detail. Quantitatively, the determined bulk gap in phase II is ~254 ± 8 meV, slightly larger than that in phase I. In addition, there seems to exist a resonance between −100 and −200 mV in the d$I$/d$V$ spectrum taken at the terrace of phase II in Fig. 3e. It is noteworthy that the band gap of the epitaxial ZrTe₅ monolayer is much larger than that measured on the cleaved surface of bulk ZrTe₅[21,51]. Compared with the 3D weak topological nature of bulk ZrTe₅, the ZrTe₅ monolayer exhibits the true 2D nature of the topological insulator in the monolayer limit.

To understand the electronic and topological properties of the ZrTe₅ monolayer in phases I and II, we further performed the density functional theory (DFT) calculations for both phases. Figure 4a, b plots the orbital-decomposed band structures of phase I without and with the SOC along four time-reversal invariant points in the first Brillouin zone (the schematic illustration of the first Brillouin zone can be found in Supplementary Fig. 11). As shown in Fig. 4a without the SOC, the energy bands of phase I near the Fermi level are mainly contributed from the Te$^d$, Te$^z$, Te$^a$ atomic $p$-orbitals, and Zr atomic $d$-orbitals in '22'-type prisms (denoted as '22'-Te$^d$-$p_y$, '22'-Te$^z$-$p_x$/'22'-Te$^z$-$p_y$, '22'-Te$^a$-$p_y$,

and '22'-Zr-$d_{z^2}$, respectively). Phase I exhibits one band crossing along the Γ-X direction near the Fermi level, dominated by the '22'-Zr-$d_{z^2}$ and '22'-Te$^a$-$p_y$, indicating the existence of inverse band orders between the Γ and X points. When the SOC is considered, as shown in Fig. 4b, the band crossing is gapped out, resulting in an insulator with a full band gap of ~169 meV. This is slightly smaller than the experimental value of ~235 ± 5 meV. Here, the role of the SOC is only to gap out the band crossing point, without inducing any extra band inversion at the high-symmetry points protected by time-reversal symmetry, which is analogous to that in graphene and bulk ZrTe₅[6,15]. In addition, due to the breaking of spatial inversion symmetry, the band structure exhibits considerable spin−orbit splitting except at the four time-reversal invariant points.

To gain further insight into the topological properties of phase I, we evaluate the $\mathbb{Z}_2$ topological invariant by tracing the evolution of Wannier charge centers using the maximally localized Wannier functions[52,53]. Based on the calculated Wilson loop (see Supplementary Fig. 14 for more details), phase I is identified to be topologically nontrivial with $\mathbb{Z}_2 = 1$. Since the existence of gapless edge states is a hallmark of 2D QSH insulators, the edge states of a semi-infinite slab perpendicular to the prism stripe (namely, perpendicular to the $a$-axis) are shown in Fig. 4c. These edge states exhibit a Dirac nature at the Ȳ point within the band gap, further demonstrating that phase I is topologically nontrivial. Moreover, since the calculated local density of states (LDOS) can make an easy comparison with the STS data[54], we calculated the LDOSs of phase I and its topological edge states in a semi-infinite nanoribbon perpendicular to the $a$-axis. As seen in Fig. 4d, the LDOS of the edge states exhibits a gapless 'V'-shaped feature within

the bulk band gap, which is well consistent with the experimental results shown in Fig. 3b.

Figure 4e, f plots the orbital-decomposed band structures of phase II without and with the SOC, respectively. As phase II has a space group of $P_{bmc}$, the '22'-type prisms and '11'-type prisms in this '2211' configuration are inversion-symmetrically equivalent. Therefore, the bands near the Fermi level are mainly contributed by the $Te^a$-$p_y$, $Te^d$-$p_y$, $Te^z$-$p_x$, $Te^z$-$p_y$, and $Zr$-$d_{z^2}$ orbitals of both the '22'-type prisms and '11'-type prisms. Similar to phase I, the band structure of phase II without the SOC in Fig. 4e shows a band crossing along the Γ-X direction, primarily occupied by the $Te^a$-$p_y$ and $Zr$-$d_{z^2}$ orbitals. When the SOC is included, as shown in Fig. 4f, this band crossing is gapped out, without inducing any extra band inversion at the high-symmetry points. Here, it is noted that the difference between phases I and II is that the symmetry of phase II prevents spin−orbit splitting along certain specific directions (such as Γ-X) with the inclusion of the SOC [see Fig. 4b, f]. In addition, we found that the nearly flat Rashba bands along the Y'-Γ-Y'' direction in the whole Brillouin zone contribute to a pronounced density of states near the energy window of [− 200 meV, − 100 meV], which may result in the experimentally observed resonance, as shown in Fig. 3e (details can be seen in Supplementary Fig. 15). We further calculated the Wilson loop of phase II (see Supplementary Fig. 16) and confirmed that it is topologically nontrivial with $\mathbb{Z}_2 = 1$. As another manifestation of nontrivial band topology, the edge states of a semi-infinite slab perpendicular to the prism stripe are shown in Fig. 4g, with the Dirac nature at the Ȳ point within the band gap, also suggesting that phase II is a QSH insulator. As shown in Fig. 4h, the calculated LDOS of the topological edge states also exhibits a gapless 'V'-shaped feature within the bulk band gap, well consistent with the experimental results shown in Fig. 3f.

In addition, we have carried out calculations using the HSE06 hybrid functional[55] to further examine the electronic and topological properties of both phases (see details in Supplementary Fig. 17). The band gap of phase I remains nearly unchanged at ~160 meV, while the band gap of phase II increases to ~157 meV. Both phases are verified to preserve their topologically nontrivial nature with $\mathbb{Z}_2 = 1$ and exhibit topologically protected edge states.

## Discussion

In summary, we have grown an epitaxial ZrTe₅ monolayer as a QSH material. The ZrTe₅ monolayer exhibits a large SOC full gap of ~254 meV and robust topological edge states localized at the periphery. Such a large band gap can efficiently suppress bulk conduction, making the ZrTe₅ monolayer essentially promising for realizing the high-temperature QSH effect. The weak vdW coupling between the ZrTe₅ monolayer and BLG/SiC substrate results in the formation of a quasi-freestanding monolayer, and thus, sample transfer is expected to be practically feasible for device construction. The experimental success of ZrTe₅ monolayers provides a highly desirable material candidate for further exploring exotic 2D topological physics.

## Methods
### Sample synthesis
The single-layer ZrTe₅ films were grown on a 4*H*-SiC(0001) substrate in a molecular beam epitaxy (MBE) system (GC-MBE-STM-UHV-0100) and a combined MBE-STM system (Unisoku, USM1500). The base pressure of the ultrahigh vacuum is $1 \times 10^{-10}$ mbar. Before ZrTe₅ epitaxy, the 4*H*-SiC(0001) substrates were degassed overnight at ~650 °C and then flashed up to ~1450 °C for a few cycles, until the surface was fully covered by bilayer graphene (BLG). During the ZrTe₅ epitaxy, the BLG/SiC substrate was kept within a rather narrow temperature window (~150−200 °C). High purity Zr (99.95%) and Te (99.999%) were co-evaporated from an electron beam evaporator and a Knudsen effusion cell, respectively. The ratio of Zr:Te flux was carefully set to ~1:5 to 1:10, to provide sufficient reactive Te atoms and simultaneously avoid the

abundant Te condensation on the surface. The growth process was in situ monitored by reflection high-energy electron diffraction (RHEED).

### STM/STS characterization
After epitaxy, the sample was transferred to a low-temperature STM (Unisoku, USM1500) for the scan. All the STM and STS measurements were carried out at ~4.5 K unless otherwise specified. The STM images were taken under a constant current mode. The STS spectra were collected by using a lock-in technique with an ac modulation of ~5 – 12 mV at 879 Hz.

### X-ray photoelectron spectroscopy measurements
XPS measurements were carried out in an ultra-high vacuum (UHV, ~$5 \times 10^{-10}$ mbar) chamber, equipped with a hemispherical electron energy analyzer (ESCALAB Xi+, Thermo) and a monochromatic Al K$_\alpha$ X-ray source of 1486.7 eV. X-ray with a spot size of 100 μm diameter was adopted during high-resolution XPS spectra measurement. Before XPS measurement, the sample was in situ annealed at 190 °C for one hour to eliminate the Te capping layer, leaving clean ZrTe₅ surface. The binding energy (BE) of core-level peaks was calibrated concerning the C-C 1*s* bond (BE = 284.8 eV). After subtracting a Shirley-type background, the spectra were curve-fitted.

### Determination of the band gap from STM spectra
The size of band gaps was determined by a statistical analysis of the scanning tunneling spectroscopy d*I*/d*V* data taken far away from the step edge. We first did the logarithmic operation to the spectra, and then obtained the intersection of the exponential conduction and valence band tails with the noise baseline, as the conduction band minimum (CBM) and valence band maximum (VBM), respectively. The band gap was defined by the difference of the VBM and CBM. The reported values of ~235 ± 5 meV for phase I (~254 ± 8 meV for phase II) were obtained by averaging the band gap values of ~25 individual d*I*/d*V* curves for each phase, and the uncertainty was determined by adding the standard deviations of the means.

### DFT calculations
The first-principles calculations were performed with density functional theory (DFT) implemented in the Vienna ab initio simulation package (VASP)[56,57], using the generalized gradient approximation (GGA) of Perdew, Burke and Ernzerhof (PBE)[58] as the exchange-correlation functional. For VASP calculations, we used a plane-wave cutoff energy of 500 eV and a $15 \times 1 \times 3$ Monkhorst-Pack *k* mesh[59] in the first Brillouin zone. A vacuum layer of more than 15 Å was adopted to ensure decoupling between neighboring slab images. All the atoms were allowed to fully relax during structural optimization until all the forces on each atom were less than 0.01 eV/Å. The core electrons were treated fully relativistically, and valence electrons were treated in a scalar relativistic approximation. The phonon dispersion relations were obtained by adopting the supercell approach with the finite displacement method[47]. The $\mathbb{Z}_2$ invariants were calculated via the Wannier charge centers[60] and nontrivial edge states were obtained from maximally localized Wannier functions as implemented in the Wannier90 and WannierTools packages[52,53].

## Data availability
All data that support the findings of this study are present in the paper and the Supplementary Information. Further information can be acquired from the corresponding authors upon request.

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

## Acknowledgements

This work was financially supported by the National Key Research and Development Program of China (Grants No. 2021YFA1400403), the National Natural Science Foundation of China (Grants Nos. 12374183, 92165205, 12374458, 12004368, and 11974323), the Innovation Program for Quantum Science and Technology (Grant No. 2021ZD0302800), the Strategic Priority Research Program of Chinese Academy of Sciences (Grant No. XDB0510200), the Anhui Initiative in Quantum Information Technologies (Grant No. AHY170000), and the Suzhou Science and Technology Program (SJC2021009).

## Author contributions

S.C. L. conceived the project. Y.J. X. grew the epitaxial ZrTe$_5$ monolayers and performed the STM characterizations with the assistance of Q.Y. L., C.L. X., W.M. Z., Q.W. W., L.G. D., X. D., Y.X. M., Y.K. W., Y.H. G., and Z.Y. J. S.C. L. and D.X. supervised the sample growth and experimental characterizations. G.C. performed the first-principles DFT calculations under the supervision of P.C. and Z.Z. W.L., L.J., and F.S.L. performed the XPS measurements. Y.J.X., G.C., P.C., Z.Z., and S.C.L. wrote the manuscript with inputs and comments from all the authors.

## Competing interests

The authors declare no competing interests.
