## [Peer Review File · Nature Communications]

Realization of monolayer ZrTe₅ topological insulators with wide band gapsEditorial Note: Parts of this Peer Review File have been redacted as indicated to remove third-party material where no permission to publish could be obtained. When text is deleted in rebuttals and referee reports, add "[redacted]" in that location.

Reviewers' Comments:

Reviewer #1:

Remarks to the Author:

The authors present results of the STM studies of ZrTe5 2D topological insulator grown by molecular beam epitaxy as ultra-thin (1 Monolayer thick) layers on bilayer graphene / SiC substrates.

Even though such studies are interesting, especially since the reports on MBE grown ZrTe5 are scarce the authors ignore or are unaware of the results of similar STM/STS studies published already for vacuum cleaved bulk ZrTe5 and for the MBE layers:

[1] R. Wu, et. al. Evidence for Topological Edge States in a Large Energy Gap near the Step Edges on the Surface of ZrTe5, Phys. Rev. X 6, 021017 (2016)]

and for the MBE-grown layers

[2] Y. Song, et. al., Signatures of the exciton gas phase and its condensation in monolayer 1T-ZrTe2. Nat. Commun. 14, 116 (2023)].

In paper [1] concerning ZrTe5 bulk crystals cleaved in ultra high vacuum, conducting states with a nearly constant DOS inside the gap were detected near the step edge of one monolayer height; similar to the results shown in the reviewed manuscript.

Paper [2] reports the MBE growth of ZrTe5 on bilayer graphene. and focuses on charge density wave states studied by ARPES.

Even though the authors of Ref [2] concentrate on other properties of ZrTe5 (charge density waves), the MBE growth is much better documented showing RHEED patterns and large scale surface morphology images.

Several issues have to be clarified by the authors in order to make their manuscript suitable for publication in Nature Communications

(1) Discussion of their result with above mentioned references [1] and [2]

(2) More details concerning the MBE growth - RHEED images, images showing surface morphology at a larger scale.

Ad.2 - one of the important concerns is the very low substrate temperature used by the authors for ZrTe5 deposition (150 - 200 C). Such low substrate temperatures can cause formation of crystalline Te parallel to ZrTe5 since excess Te does not reevaporate at such low temperatures. The authors should comment on that.

(btw in Methods section the authors erroneously name Knudsen effusion cell, as Knudsen DIFFUSION cell...)

General remark. The paper is very short. Most likely some of the results placed in Supplementary Material can be moved to the main paper.

Concerning the former one - the XPS data shown in Fig. S1 are weird. Concerning Zr 3d peaks, the intensity of Zn 3d 5/2 is normally higher than the Zr 3d 3/2, Probably the peaks are erroneously labelled. Moreover the small bumps at higher energies with respect to the main peaks of Zr and Te core levels can be due to oxidation, why?

To conclude, the authors should address above mentioned remarks, and stress novelty of their paper with respect to Refs. [1] and [2].

Reviewer #2:

Remarks to the Author:

The paper by Xue et al. reports on the successful van-der-Waals epitaxy growth of ZrTe5 monolayers on graphene/SiC substrates. Scanning tunnelling spectroscopy at 4K detects 2D topologically insulating behaviour as reflected in a bulk energy gap and metallic edge states. Overall, I find the

paper interesting, and the data and conclusions convincing. However, several major amendments need to be made in terms of analysis, presentation and addition of supporting data, before I can recommend publication.

The quality of the crystal growth and spectroscopy is somewhat limited. However, I can look beyond this as vdW crystal growth of atomic monolayers - especially at the low temperature explored in the paper - is very challenging. Given that this is the first-time demonstration of successful monolayer growth, the results are well-worth publishing.

My main concern lies more with the spectral analysis of the material, especially with the claim of "wide" band gaps (as it is read in the title of the paper). The claim that is currently not well supported by the data analysis, and the presentation of the data lacks, as per my comments below:

Major comments:

1. Line 149 and Fig. 3 a,e: "A full gap of as large as $\sim 220\text{mV}$ is identified [...]". I am not convinced of the extraction of the band gap magnitude. All data in Fig.3a,e appears bunched to the x-axis, which makes it impossible to get a true sense of the measured gap - especially whether it is indeed a "full gap" (i.e. hard gap with vanishing LDOS within the gap), or not, and what the gap's magnitude is. I suggest twofold, to

(a) plot the data on a linear scale as shown but zoom in on the y-axis (e.g. between 0 and 3 a.u.), to make the amount of gapping more clearly visible, and

(b) plot the same data on a semilogarithmic scale, as insert or in a separate panel. The latter will allow to identify the exponential band tails and where they intersect with the noise floor of the measurement. The difference in energy between the intersection points for conduction and valence bands is often taken as a more reliable measure of the bandgap (see. e.g. Fig. 2 of Nat. Mater. 13, 1091 (2014)).

Unless a more rigorous measure of the bandgap (with confidence bounds) is presented, I cannot recommend publication.

2. Fig.3b,f: similar comment as above: Zoom-in on the y-axis for clarity, such that data quality and spectral detail can be clearly assessed. Please also include a spectrum of the graphene substrate, taken at some distance from the edge to highlight differences in the spectral detail of the edge state.

3. Fig.1a shows some brighter contrast regions within the ZrTe₅ monolayer islands. I assume that these are bilayer regions? How do spectra in the bilayer regions differ from those of monolayers, e.g. in terms of the bulk gap? Do bilayers have inversion symmetry in either phase? If not, would a gap be expected to be topologically trivial or non-trivial? Do the bilayer domains have edge states?

4. Fig.4: The calculated LDOS, comparing bulk and edge, should be included here as additional panels such that easy comparison can be made with the STS data in Fig.3b,f. A good example can be found e.g. in Phys. Rev. B 105, 094512 (2022).

5. Methods: The paper claims "elegant control" of the van-der-Waals epitaxy. It is hard to assess where the "elegance" comes in, given that not much process detail has been given. From line 69, the paper claim that "ZrTe₅ monolayers can only be successfully obtained within the rather narrow windows of temperature and flux ratio." A temperature window is quoted in the Methods section, but I could not find any information on the flux ratio. It would also be good to see additional STM data in the supplementary information, showing how different flux ratios and temperature affect the growth.

6. Differences in appearance of lattice structure often emerge due to interactions with a substrate.

This can arise due to a mismatch of lattice constants and rotational alignment, e.g. in the presence of a Moire superstructure. How can the reader be ensured that the two crystal "phases" are not just due to substrate interactions, e.g. for domains that are differently rotated with respect to the substrate's lattice, rather than actually different lattices?

Minor comments:

line 61: "exotic phenomena" - be explicit what they are and list, ideally including references

line 65: "exfoliation" refers to mechanical isolation of layers from bulk crystals. I believe, in the context of the paragraph, this should read "synthesis" instead

line 65: "freestanding" should probably read "quasi-freestanding", given that the material is grown on a graphene substrate. A reference could be given, which shows that QSH insulators on graphene are quasi-freestanding in terms of their electronic structure.

Fig.3e: What is the reason for the resonance observed at -100meV in Fig.3e? e.g. in comparison to the bandstructure calculations?

Fig.3d,h: suggest to include exponential fits to the spatially decaying LDOS profiles. I understand that data quality is limited, but guides to the eye would still be helpful illustrate the energy-dependence of the imaginary wave vector for states decaying into the vacuum of the bulk (at least in Fig.3h).

Once again, I believe that this local probe spectroscopy characterization of the QSH candidate ZrTe5 is well-worth publishing in Nature Communication, once all the above comments are successfully addressed.

Reviewer #3:

Remarks to the Author:

This work is the first to achieve monolayer ZrTe5, which is of some importance. However, quantum spin Hall insulators have been well studied in the last 20 years, and there is no new physics in this work. On the other hand, Bi on SiC substrate has already been found to be a quantum spin Hall insulator with a band gap of 800 meV, which is much larger than the band gap of monolayer-ZrTe5. Therefore, the implementation of the present experiment has not exceeded the previous works. Therefore, in terms of the general interest, the impact and novelty, this work does not meet the high criteria of Nature Communications. Nevertheless, it might be suitable for specified journals. Here I raise two questions on the present manuscript even for its submission to other specified journals.

(1) The calculated energy gaps of two different phases of monolayer ZrTe5 are both smaller than the experimental value, which should be due to the underestimation of the energy gap by PBE type exchange correlation functional. The electronic structure of monolayer ZrTe5 can be calculated by hybrid functional or MBJ method, which may give rise to the results being consistent with the experiment.

(2)The MBE-grown monolayer ZrTe5 is different from the single-layer structure of three-dimensional bulk materials. And two new phases were obtained in monolayer ZrTe5, reported in the manuscript. Whether it is possible to obtain more new phases, resulting in quantum spin Hall insulators with larger energy gaps in monolayer ZrTe5?

Responses to Reviewer Comments

We thank the three reviewers for their expertized assessments of the above manuscript, and their valuable comments and suggestions. Reviewers #1 and #2 also judge that the work is suitable for publication in Nature Communications after proper revision. In this response, we have carefully addressed the questions and issues from all the reviewers, and revised the manuscript accordingly. We hope that the revised manuscript is now suitable for publication in Nature Communications.

Detailed responses are as follows.

Reviewer #1 (Remarks to the Author):

The authors present results of the STM studies of ZrTe₅ 2D topological insulator grown by molecular beam epitaxy as ultra-thin (1 Monolayer thick) layers on bilayer graphene / SiC substrates. Even though such studies are interesting, especially since the reports on MBE grown ZrTe₅ are scarce the authors ignore or are unaware of the results of similar STM/STS studies published already for vacuum cleaved bulk ZrTe₅ and for the MBE layers:

[1] R. Wu, et. al. Evidence for Topological Edge States in a Large Energy Gap near the Step Edges on the Surface of ZrTe₅, Phys. Rev. X 6, 021017 (2016)]

and for the MBE-grown layers

[2] Y. Song, et. al., Signatures of the exciton gas phase and its condensation in monolayer 1T-ZrTe₂. Nat. Commun. 14, 116 (2023)].

In paper [1] concerning ZrTe₅ bulk crystals cleaved in ultra high vacuum, conducting states with a nearly constant DOS inside the gap were detected near the step edge of one monolayer height; similar to the results shown in the reviewed manuscript.

Paper [2] reports the MBE growth of ZrTe₅ on bilayer graphene. and focuses on charge density wave states studied by ARPES. Even though the authors of Ref [2] concentrate on other properties of ZrTe₅ (charge density waves), the MBE growth is much better documented showing RHEED patterns and large scale surface morphology images.

Response: We thank the reviewer again for the positive comments. We also appreciate the reviewer for bringing the two relevant references to our attention. Our point-to-point responses to the reviewer's questions are given below.

Several issues have to be clarified by the authors in order to make their manuscript suitable for publication in Nature Communications

(1) Discussion of their result with above mentioned references [1] and [2]

Response: Regarding the first paper [R. Wu et al., Phys. Rev. X 6, 021017 (2016)] the

reviewer mentioned, our group has in fact independently published a highly related work essentially at the same time [see Ref. 21 in the original submission, X.-B. Li et al. PRL 116, 176803 (2016)]. Both the PRX and PRL papers reported the existence of the band gap in bulk ZrTe₅ and the topological edge state at the cleaved surface, and thus claimed that the bulk ZrTe₅ is a weak 3D topological insulator. Here we wish to emphasize that the novelty of our present work is to establish that the as-grown ZrTe₅ monolayer is a true 2D topological insulator, achieved for the first time in the monolayer limit, instead of a 3D topological nature for bulk ZrTe₅. Moreover, the gap of ~220 mV is much larger than that (~ <100 mV) in the bulk ZrTe₅.

We also wish to respectively point out that the second paper [Y. Song et al., Nat. Commun. 14, 116 (2023)] reported the MBE-grown ZrTe₂ monolayer, rather than the ZrTe₅ monolayer. The ZrTe₂ monolayer possesses a common structure of transition metal dichalcogenides (TMDs), where two Te layers sandwich a Zr layer with octahedral coordination (1T phase), and exhibits the charge density waves as many TMD monolayers do. In fact, prior to that NC, our group had also reported another work about the MBE-grown ZrTe₂ monolayer [L.-N. Yang et al., Appl. Phys. Lett. 120, 073105 (2022)]. Compared with the ZrTe₅ monolayer, the MBE growth of ZrTe₂ monolayer is technically less challenging, since ZrTe₂ is a thermodynamically more stable phase.

In the revised manuscript, we have added the first mentioned paper as new Ref. [51], and added a short discussion with the work of cleaved bulk ZrTe₅. Specifically, on page 8, we have added “*It is noteworthy that the band gap of the epitaxial ZrTe₅ monolayer is much larger than that as measured on the cleaved surface of bulk ZrTe₅ [21,51]. Compared with the 3D weak topological nature of bulk ZrTe₅, the ZrTe₅ monolayer exhibits the true 2D nature of the topological insulator in the monolayer limit*”. Furthermore, we have cited the second mentioned paper and our own work on the MBE-grown ZrTe₂ monolayers as new Refs. [43,44], and added “*As reference systems, the epitaxial ZrTe₂ monolayer has been investigated in previous studies [43,44]*” on page 4.

In short, we appreciate that the reviewer’s comments have helped us to give a broader perspective on the topic.

(2) More details concerning the MBE growth - RHEED images, images showing surface morphology at a larger scale.

Response: We thank the reviewer for this suggestion that prompts us to further clarify on the detailed aspects of growth. Figure R1 below shows the sequential RHEED patterns obtained during the growth of ZrTe₅ monolayers, as well as the larger-scale STM images (400 × 400 nm²).

Fig. R1 Epitaxial ZrTe_5 monolayers grown on the BLG/SiC(0001) substrate. (a) RHEED patterns collected on the bare bilayer graphene (BLG)/SiC substrate (top panel) and epitaxial ZrTe_5 monolayer with the coverage of ~ 1.0 ML (bottom panel). (b,c,d) Large-scale STM images ($400 \times 400 \text{ nm}^2$) taken on the surfaces of ZrTe_5 monolayers on the BLG/SiC substrate at various coverages. The coverages are ~ 0.3 ML for (b), ~ 0.5 ML for (c) and ~ 0.8 ML for (d), respectively. Bias voltage $U = +1.0$ V, tunneling current $I_t = 20$ pA.

In the revised manuscript, we have added the RHEED patterns and larger scale STM images as new Supplementary Fig. S1.

Ad.2 - one of the important concerns is the very low substrate temperature used by the authors for ZrTe_5 deposition (150 - 200 C). Such low substrate temperatures can cause formation of crystalline Te parallel to ZrTe_5 since excess Te does not reevaporate at such low temperatures. The authors should comment on that.

(btw in Methods section the authors erroneously name Knudsen effusion cell, as Knudsen DIFFUSION cell...)

Response: We thank the reviewer for raising this concern. We agree that growth at low substrate temperature can cause the formation of crystalline Te. The growth of pure ZrTe_5 monolayer can only be successful if the substrate temperature is set to be slightly higher than that of Te condensation. Figures R2 (a)-(c) show the large-scale surface morphologies ($200 \times 200 \text{ nm}^2$) of the ZrTe_5 monolayers grown with a constant Zr:Te flux ratio and at different substrate temperatures of ~ 120 °C, ~ 150 °C, and ~ 180 °C, respectively. It can be seen that the coexistence of ZrTe_5 monolayer and crystalline Te films occurs at ~ 120 °C, but the pure ZrTe_5 monolayers (without Te films) were obtained

at $\sim 150^\circ\text{C}$ and $\sim 180^\circ\text{C}$. On the other hand, if the substrate temperature is set to be higher than the narrow temperature window, other more stable phases such as ZrTe_3 and even ZrTe_2 monolayers (as discussed earlier) are formed instead of the ZrTe_5 monolayer. The ratio of Zr:Te flux was carefully set to $\sim 1:5$ to $1:10$, in order to provide sufficient reactive Te atoms and avoid the simultaneous Te condensation.

Fig. R2 Epitaxial ZrTe_5 monolayers grown at different temperatures. (a-c) Surface topographic image of the single-layer ZrTe_5 on the BLG/SiC substrate at $\sim 120^\circ\text{C}$, $\sim 150^\circ\text{C}$, and $\sim 180^\circ\text{C}$ ($200 \times 200 \text{ nm}^2$). Bias voltage $U = +1.0 \text{ V}$, tunneling current $I_t = 20 \text{ pA}$.

On page 4 of the revised manuscript, we have added “*In order to successfully obtain the epitaxial of ZrTe_5 monolayer, fine tuning of the epitaxy parameters is found to be crucial. The substrate temperature for ZrTe_5 growth has to be set to slightly higher than that for Te crystallization on the surface to avoid the formation of redundant Te islands, and below those for ZrTe_3 and ZrTe_2 crystallization to avoid the formation of ZrTe_3 and ZrTe_2 monolayers*”.

In the revised Methods section, we have added “*The ratio of Zr:Te flux was carefully set to $\sim 1:5$ to $1:10$, in order to provide sufficient reactive Te atoms and simultaneously avoid the abundant Te condensation on the surface*”. We have also corrected the “Knudsen diffusion cell” with “Knudsen effusion cell”.

In the Supplementary Materials, we have added Fig. R2 as a new panel in the revised Fig. S2 to include the STM morphologies obtained at different substrate temperatures.

General remarque. The paper is very short. Most likely some of the results placed in Supplementary Material can be moved to the main paper.

Response: We thank the reviewer for this suggestion. In the revised manuscript, we have added the new XPS results and the calculated local density of states (LDOS) of bulk and edge states of both phases into Fig. 1 and Fig. 4, respectively. We have also added new calculations using the HSE06 hybrid functional, new results on ZrTe_5 bilayers, as well as more experimental data of STM and RHEED in the Supplementary

Materials.

Concerning the former one - the XPS data shown in Fig. S1 are weird. Concerning Zr 3d peaks, the intensity of Zn 3d 5/2 is normally higher than the Zr 3d 3/2, Probably the peaks are erroneously labelled. Moreover the small bumps at higher energies with respect to the main peaks of Zr and Te core levels can be due to oxidation, why?

Response: We thank the reviewer for these comments. The XPS measurements were *ex situ* performed in a separate chamber. We believe the higher energy bumps, as well as the “abnormal” intensities of Zr 3d_{5/2} and 3d_{3/2}, are due to the oxidation during sample transfer from the ultrahigh vacuum (UHV) chamber to air. In order to avoid oxidation during transfer, we have capped the sample with Te protection layers and re-performed the XPS measurement. Prior to the XPS measurement, the sample was annealed in UHV to remove the capping layers. Below in Figure R3 we show the new XPS results, where one can find the binding energies (180.57 eV for Zr 3d_{5/2} and 182.95 eV for Zr 3d_{3/2}) and peak intensity ratio are both consistent with those of the bulk ZrTe₅ as reported in previous literatures [Nano Lett. 16, 7364-7369 (2016) and J Raman Spectrosc. 53, 104–112 (2022)].

[Redacted]

Fig. R3 X-ray photoelectron spectroscopy data taken on the epitaxial ZrTe₅ monolayers. (a,b) Our new XPS results of Zr 3d and Te 3d peaks. (c,d) XPS data on bulk ZrTe₅ extracted from literatures [(c): Nano Lett. 16, 7364-7369 (2016); (d): J. Raman Spectrosc. 53:104–112 (2022)].

In the revised manuscript, we have added the new XPS results shown in Figs. R3(a) and (b) to Fig. 1, and added the corresponding descriptions of the XPS measurements to the Methods section.

To conclude, the authors should address above mentioned remarks, and stress novelty of their paper with respect to Refs. [1] and [2].

Response: We hope we have provided adequate response to the reviewer's questions, and the manuscript in the current format is now suitable to be published in Nature Communications.

Reviewer #2 (Remarks to the Author):

The paper by Xu et al. reports on the successful van-der-Waals epitaxy growth of ZrTe₅ monolayers on graphene/SiC substrates. Scanning tunnelling spectroscopy at 4K detects 2D topologically insulating behaviour as reflected in a bulk energy gap and metallic edge states. Overall, I find the paper interesting, and the data and conclusions convincing. However, several major amendments need to be made in terms of analysis, presentation and addition of supporting data, before I can recommend publication.

The quality of the crystal growth and spectroscopy is somewhat limited. However, I can look beyond this as vdW crystal growth of atomic monolayers - especially at the low temperature explored in the paper - is very challenging. Given that this is the first-time demonstration of successful monolayer growth, the results are well-worth publishing.

Response: We thank the reviewer for the positive comment on our work. Our point-to-point responses to the reviewer's questions are given below.

My main concern lies more with the spectral analysis of the material, especially with the claim of "wide" band gaps (as it is read in the title of the paper). The claim that is currently not well supported by the data analysis, and the presentation of the data lacks, as per my comments below:

Major comments:

1. Line 149 and Fig. 3 a,e: "A full gap of as large as ~ 220 mV is identified [...]". I am not convinced of the extraction of the band gap magnitude. All data in Fig.3a,e appears bunched to the x-axis, which makes it impossible to get a true sense of the measured gap - especially whether it is indeed a "full gap" (i.e. hard gap with vanishing LDOS within the gap), or not, and what the gap's magnitude is. I suggest twofold, to

(a) plot the data on a linear scale as shown but zoom in on the y-axis (e.g. between 0 and 3 a.u.), to make the amount of gapping more clearly visible, and

(b) plot the same data on a semilogarithmic scale, as insert or in a separate panel. The latter will allow to identify the exponential band tails and where they intersect with the noise floor of the measurement. The difference in energy between the intersection points for conduction and valence bands is often taken as a more reliable measure of the bandgap (see. e.g. Fig. 2 of Nat. Mater. 13, 1091 (2014)).

Unless a more rigorous measure of the bandgap (with confidence bounds) is presented, I cannot recommend publication.

Response: We thank the reviewer for this insightful and invaluable suggestion. Following the reviewer's suggestion, we have plotted one of the original STS spectra on a zoom-in y-scale between 0 and 3 a.u. for both phases, as shown in Figures R4(a) and (b). Both phases I and II show the full gaps with vanishing LDOS within the gaps. We also replotted the same data on a semilogarithmic scale of y-axis, as shown in

Figures 4R(c) and (d). The exponential band tails and where they intersect with the noise floor of the measurement are identified, as marked in Figure 4R(c) and (d). We then followed the previous literature and took the difference in energy between the intersection points as a measure of the bandgap, which was determined to be ~ 235 mV for phase I and ~ 254 mV for phase II.

Fig. R4 Differential conductance dI/dV spectra taken on the (a) phases-I and (b) phase-II terraces away from the step edges ($U = +500$ mV, $I_t = 200$ pA, $U_{\text{mod}} = 7$ mV). (c, d) The same data of (a) and (b) replotted on a semilogarithmic scale of y-axis and a smaller linear scale of x-axis.

In the revise manuscript, we have replaced Figs. 3(a) and (e) with the new version of Figs. R4(a)-(d), and updated the figure captions. The original Figs. 3(a) and (e) were moved to the Supplementary Materials as new Fig. S13. On page 8, we have added “Taking the difference in energy between the intersection points for conduction and valence bands [Ref: *Nat. Mater.* 13, 1091 (2014)]”. We have also updated the band gaps for phases I and II to 235 mV and 254 mV, respectively throughout the manuscript.

2. Fig.3b,f: similar comment as above: Zoom-in on the y-axis for clarity, such that data quality and spectral detail can be clearly assessed. Please also include a spectrum of the graphene substrate, taken at some distance from the edge to highlight differences in the spectral detail of the edge state.

Response: As shown in Figure R5 below, we replotted the dI/dV spectra in Fig. 3(b,f) for better clarity, and also included a dI/dV spectrum taken on the graphene substrate

to highlight the differences in the spectral details of the edge states.

Fig. R5 (a) Comparison of three representative dI/dV spectra taken at the phase-I terrace (black), at the step edge (red) and at the graphene substrate far from the step edge (blue). $U = +500$ mV, $I_t = 200$ pA, $U_{\text{mod}} = 7$ mV. (b) Same as (a) but for phase II.

In the revised manuscript, we have replaced the Figs. 3(b) and 3(f) with Fig. R5 and updated the captions.

In short, these comments on the data analyses and presentations have substantially improved the clarity and validity of the claims on the 2DTI gaps.

3. Fig.1a shows some brighter contrast regions within the ZrTe_5 monolayer islands. I assume that these are bilayer regions? How do spectra in the bilayer regions differ from those of monolayers, e.g. in terms of the bulk gap? Do bilayers have inversion symmetry in either phase? If not, would a gap be expected to be topologically trivial or non-trivial? Do the bilayer domains have edge states?

Response: The brighter contrast regions are indeed the relatively small ZrTe_5 bilayer regions. Similar to the first layer, the second layer of ZrTe_5 also mainly exhibits both phases-I and II structures, but, different from the first layer, the phase of bulk ZrTe_5 is also observable in the second layer, as shown in Fig. R6. In order to respond to the reviewer's questions, we have taken the dI/dV spectra in these bilayer regions, both at the center and step edge, as shown in Fig. R6. The bandgaps of the second-layer regions are identified to be ~ 30 meV for phase I, ~ 90 meV for phase II, and ~ 80 meV for bulk phase, all of which are substantially narrower than that of the ZrTe_5 monolayer. At the step edge, there is a finite LDOS within the bandgap as well, which is similar to the case of ZrTe_5 monolayers. However, the edge state at the bilayer is not so robust as the

first layer, i.e., there exist some regions at the step edge, where the edge state is rather weak, as shown in Fig. R6(f). Considering the small size (~ 5 to 10 nm in width) of the second-layer ZrTe_5 islands, this is possibly due to the confinement of the limited size of bilayers, or the coupling between the edge states.

Fig. R6 (a)-(c) STM images (3d view: $25 \times 25 \text{ nm}^2$) showing the bilayer ZrTe_5 islands with the second layer in phase I (a), phase II (b) and bulk phase (c), respectively. The yellow rectangles mark the second-layer ZrTe_5 regions, and underneath the second-layer regions are the first-layer ZrTe_5 . The black and red dots mark the locations where the dI/dV spectra are taken. (d) dI/dV spectra taken at the center (black) and edge (red) of the second-layer ZrTe_5 in phase I. For comparison, the dI/dV spectrum taken at the first-layer ZrTe_5 of phase I is also plotted (gray). (e, f) dI/dV spectra taken on the second-layer ZrTe_5 in phase II and bulk phase, respectively.

In order to gain further insight into the electronic and topological properties of ZrTe_5 bilayers, we have performed additional first-principles calculations of three homobilayers, which are respectively the most stable configurations in bulk phase, phase I, and phase II, with AA, AA, and AB stacking, respectively. The crystal structures, band structures, Wilson loops, and edge states of the three homobilayers are shown in Figs. R7-R9, and the corresponding band gaps, space group, symmetry, and topological invariants Z_2 are summarized in Table RI. The band gaps of the three homobilayers are calculated to be 91, 46, and 50 meV, respectively, all of which are narrower than those of the ZrTe_5 monolayers (phases I and II). The bilayers in bulk phase and phase II exhibit inversion symmetry, while the bilayer in phase I lacks this symmetry. The three bilayers are all identified to be topologically trivial with $Z_2 = 0$, with trivial edge states displayed in Figs. R7(c), 8(c), and 9(c). Therefore, such edge

states do not exhibit topologically protected nature and are not robust. These findings are qualitatively (and semi-quantitatively) consistent with the experimentally observed band gaps and weak edge states of the ZrTe₅ bilayers. It is worth cautioning that, due to significant lattice mismatches between phase I and the other two phases, we did not consider heterobilayers that may exist in experiments, and our current calculations cannot rule out the presence of topological edge states in the case of heterobilayers (namely, the upper and lower layers consist of different phases of I and II and the system is stabilized by the substrate).

Figure R7 (a) Top (upper panel) and side (lower panel) views of the crystal structure, (b) band structures obtained from the DFT calculations and Wannier interpolation, (c) Wilson loop, and (d) edge states of the ZrTe₅ homobilayer with AA stacking in bulk phase. The black rectangle in (a) represents the primitive unit cell.

Figure R8 (a) Top (upper panel) and side (lower panel) views of the crystal structure, (c) band structure obtained from the Wannier interpolation, (b) Wilson loop, and (d) edge states of the ZrTe₅ homobilayer with AA stacking in phase I. The black primitive unit cell in (a) is used for calculations.

Figure R9 (a) Top (upper panel) and side (lower panel) views of the crystal structure, (c) band structure obtained from the Wannier interpolation, (b) Wilson loop, and (d) edge states of the ZrTe₅ homobilayer with AB stacking in phase II. The black rectangle in (a) represents the primitive unit cell.

Table RI Summarized crystal parameters, band gaps, space group, with/without inversion symmetry, and Z_2 values for the three homobilayers, which are the respective most stable configurations in bulk phase, phase I, and phase II.

Samples	a (Å)	c (Å)	$\angle aoc$ (°)	Gap (meV)	Space group	Inversion symmetry	Z_2
Bulk bilayer	4.0	13.81	90	91	Pmma	yes	0
Phase-I bilayer	4.02	20.66	95.57	46	Cm	no	0
Phase-II bilayer	4.02	27.09	90	50	P2/m	yes	0

On page 5 of the revised manuscript, we have added “*It is noteworthy that the brighter contrast regions on top of the ZrTe₅ monolayers, as shown in Fig. 1(a), are the second layer ZrTe₅ islands. Besides the phases I and II that dominate the second layer*

ZrTe₅, the bulk phase of ZrTe₅ also starts to appear in the second layer (more experimental data and first-principles calculations for the bilayers can be found in Supplementary Figs. S4-S7). Due to the much smaller sizes and complicated interfacial structures, the second layer ZrTe₅ is out of the focus of the present work.”

In the revised Supplementary Materials, we have added Figs. R6-R9 as new Fig. S4-S7, and added Table RI as new Table SII.

4. Fig.4: The calculated LDOS, comparing bulk and edge, should be included here as additional panels such that easy comparison can be made with the STS data in Fig.3b,f. A good example can be found e.g. in Phys. Rev. B 105, 094512 (2022).

Response: Following the reviewer’s suggestion, we have compared the density of states (DOS) of phase I and phase II and the corresponding local DOS (LDOS) of their edge states in semi-infinite nanoribbons perpendicular to the prism strip (namely, perpendicular to the a axis). As seen in Fig. R10, the LDOS exhibits a gapless ‘V’-shaped feature of the edge states within the bulk band gap for each phase, which is well consistent with the experimental results (see Fig. 3(b,f) in the main text).

Figure R10 (a) Topological edge states of phase I in a semi-infinite slab perpendicular to the prism edge (namely, along the c axis). The warmer colors denote a higher local density of states, and the blue regions denote the bulk band gaps. (b) Calculated local density of states (DOS) of the bulk (black) and edge (red) states of phase I. (c,d) Same as (a,b) but for phase II.

To reflect the new results of the calculated LDOS of bulk and edge states for both phases, we have added Fig. R10(b,d) as new Fig. 4(d,h) in the revised manuscript.

On page 9 of the revised manuscript, we have added “Moreover, since the calculated local density of states (LDOS) can make easy comparison with the STS data [54], we calculated the LDOSs of phase I and its topological edge states in semi-infinite nanoribbons perpendicular to the a axis. As seen in Fig. 4(d), the LDOS of the edge states exhibits a gapless ‘V’-shaped feature within the bulk band gap, which is well consistent with the experimental results shown in Fig. 3(b).”, together with new Ref. [54].

On page 10 of the revised manuscript, we have added “As shown in Figs. 4(h), the calculated LDOS of the topological edge states also exhibits a gapless ‘V’-shaped feature within the bulk band gap, well consistent with the experimental results shown in Fig. 3(f).”

5. Methods: The paper claims "elegant control" of the van-der-Waals epitaxy. It is hard to assess where the "elegance" comes in, given that not much process detail has been given. From line 69, the paper claim that "ZrTe₅ monolayers can only be successfully obtained within the rather narrow windows of temperature and flux ratio." A temperature window is quoted in the Methods section, but I could not find any information on the flux ratio. It would also be good to see additional STM data in the supplementary information, showing how different flux ratios and temperature affect the growth.

Response: We thank the reviewer for raising these technical yet crucial aspects. In the revised Methods section, we have added “The ratio of Zr:Te flux was carefully set to ~1:5 to 1:10, in order to provide sufficient reactive Te atoms and simultaneously avoid the abundant Te condensation on the surface”. In the Supplementary Materials, we have added additional STM data taken with different substrate temperatures and Zr:Te flux ratios, as new Fig. S2.

On page 4 of the revised manuscript, we have added “In order to successfully obtain the epitaxial of ZrTe₅ monolayer, fine tuning of the epitaxy parameters is found to be crucial. The substrate temperature for ZrTe₅ growth has to be set to slightly higher than that for Te crystallization on the surface to avoid the formation of redundant Te islands, and below those for ZrTe₃ and ZrTe₂ crystallization to avoid the formation of ZrTe₃ and ZrTe₂ monolayers.”.

In the revised Supplementary Materials, we have added Fig. R11 as new Fig.S2.

Fig. R11 Epitaxial ZrTe_5 monolayers grown at different substrate temperatures and flux ratios of Zr:Te. (a) Top panel: STM images ($200 \times 200 \text{ nm}^2$) of the ZrTe_5 sample on BLG/SiC grown at $\sim 120^\circ\text{C}$, $\sim 150^\circ\text{C}$, and 220°C respectively, with the Zr:Te flux ratio fixed at 1:8. Bias voltage $U = +1.0 \text{ V}$, tunneling current $I_t = 20 \text{ pA}$. Bottom panel: Zoom-in images of the top panels showing the Te island, ZrTe_5 , ZrTe_3 , and ZrTe_2 layers. (b) STM images of the ZrTe_5 on BLG/SiC grown at different Zr:Te flux ratios, with the substrate temperature fixed at $\sim 180^\circ\text{C}$. Bias voltage $U = +1.0 \text{ V}$, tunneling current $I_t = 20 \text{ pA}$.

6. Differences in appearance of lattice structure often emerge due to interactions with a substrate. This can arise due to a mismatch of lattice constants and rotational alignment, e.g. in the presence of a Moire superstructure. How can the reader be ensured that the two crystal "phases" are not just due to substrate interactions, e.g. for domains that are differently rotated with respect to the substrate's lattice, rather than actually different lattices?

Response: We thank the reviewer for this comment. Figure R12(a) below shows a large-scale STM image, in which the two phases with various orientations can be observed, as highlighted in Figs. R12(b) and (c) and the zoom-in images in the bottom. One can tell that the lattice structures of both ZrTe₅ phases are independent on their relative crystal orientations to the graphene substrate. Moreover, both phases can even coexist in one monolayer island with the same orientation (see images 1, 4, 5, and 8). These observations can sufficiently exclude the possibility that the two crystal phases are different appearances of the same lattice structure due to substrate interactions.

On page 5 of the revised manuscript, we have added “*Our large-scale STM data further demonstrate that the orientations of ZrTe₅ monolayers in both phases are randomly distributed, regardless that of the BLG/SiC substrate, further indicating a rather weak interlayer vdW interaction (more data can be found in Supplementary Fig. S3)*” and “*It is well known that the mismatch of lattice constants and rotational alignments between the epilayer and substrate, e.g., in the presence of a Moire superstructure, can give rise to a difference in the topographic appearances as well. However, the statistic of our STM data show that the morphologies of phases I and II are either independent on their lattice orientations. Moreover, both phases can even coexist in one monolayer island with the same orientation (more details can be found in Supplementary Fig. S3). Thus, it is concluded that the phases I and II of ZrTe₅ monolayers are intrinsically two distinct lattice structures, but not different appearances of the same lattice structure due to substrate interactions*”.

In the revised Supplementary Materials, we added Fig. R12 as new Fig. S3.

Fig. R12 (a) Large-scale STM image ($150 \times 150 \text{ nm}^2$) of the epitaxial ZrTe_5 monolayers on the BLG/SiC substrate. Bias voltage $U = +1.0 \text{ V}$, tunneling current $I_t = 20 \text{ pA}$. (b) Derivative image of (a). Inset: FFT image of (b) showing the orientation of the ZrTe_5 monolayers. (c) The same image of (b) with 10 labeled squares from “1” to “10”. The zoom-in images ($30 \times 30 \text{ nm}^2$) of the labeled squares are depicted in the bottom with the same labels. The phases I and II regions are marked by “I” and “II” in the zoom-in images.

Minor comments:

line 61: "exotic phenomena" - be explicit what they are and list, ideally including references

Response: In response to the reviewer’s comment, on page 3 in the main text, the sentence “*exotic phenomena have been discovered in bulk ZrTe_5 [36-42]*” has been replaced by “*exotic phenomena have been discovered in bulk ZrTe_5 , such as chiral magnetic effect, anomalous Hall effect, 3D quantum Hall effect and log-periodic oscillations, etc. [36-42]*”.

line 65: "exfoliation" refers to mechanical isolation of layers from bulk crystals. I believe, in the context of the paragraph, this should read "synthesis" instead

Response: We have replaced “*exfoliation*” with “*synthesis*” in the revised manuscript.

line 65: "freestanding" should probably read "quasi-freestanding", given that the material is grown on a graphene substrate. A reference could be given, which shows that QSH insulators on graphene are quasi-freestanding in terms of their electronic structure.

Response: We thank the reviewer for this suggestion. Previous literatures [e.g., Refs: Nat. Phys. 13, 683 (2017); PRB 96, 041108 (2017)] reported that the epitaxially grown 1T'-WTe₂ monolayer on graphene, as a QSH candidate, is quasi-freestanding in terms of its electronic structure measured by ARPES and STM.

In the revised manuscript, we have changed “*freestanding*” to “*quasi-freestanding*”. On page 3, at the end of the last sentence, and two references [Nat. Phys. 13, 683 (2017); PRB 96, 041108 (2017)] have been cited as examples of quasi-freestanding QSH monolayers on graphene.

Fig.3e: What is the reason for the resonance observed at -100meV in Fig.3e? e.g. in comparison to the band structure calculations?

Response: Following the reviewer’s suggestion, we have calculated the DOS of phase II, as shown in the right panel of Fig. R13(a). It can be seen that a resonance also appears near -100 meV in the DOS (marked by the red arrow), which is consistent with the experimental results. To understand the physical reason for the presence of the resonance, we have further examined the energy dispersions in the whole Brillouin zone. The energy dispersion relation of the highest valence band at the $k_z = 0$ plane is plotted in Fig. R13(c). It is noted that there exist Rashba-type bands along the M'-Y-M direction [Fig. R13(b)] besides the bands at the Γ point. From Figs. R13(c) and (d), these Rashba bands are nearly flat along the Y'- Γ -Y'' direction, thereby resulting in a pronounced DOS near -100 meV. As a result, such a pronounced DOS in the electronic structure contributes to the experimentally observed resonance at -100 meV.

It is noteworthy that even though the resonance always exists, its position and intensity depend sensitively on the specific tunneling conditions. The resonance is very prominent in some dI/dV spectra, while not so prominent in some other spectra. In the revised manuscript, we have replaced Fig.3(e) with the newly collected spectra.

Figure R13 (a) Band structure (left panel) and DOS (right panel) of phase II. (b) Band structure along the direction M'-Γ-M. (c) Energy dispersion relations of the highest valence band at the $k_z = 0$ plane, where the energy eigenvalues are denoted by different colors. (d) Energy dispersion relation of the highest valence band along the pink line in (c).

On page 8 of the revised manuscript, we have added “ In addition, *there seems to exist a resonance between -100 and -200 meV in the dI/dV spectrum taken at the terrace of phase II in Fig.3(e).*”

On page 10 of the revised manuscript, we have added “*In addition, we found that the nearly flat Rashba bands along the Y'-Γ-Y'' direction in the whole Brillouin zone contribute to a pronounced density of states near the energy window of [-200 meV, -100 meV], which may result in the experimentally observed resonance, as shown in Fig. 3(e) (details can be seen in Supplementary Fig. S15).*” and modified “... the symmetry of phase II prevents spin-orbit splitting with the inclusion of the SOC [see Fig. 4(b, e)].” to “... the symmetry of phase II prevents spin-orbit splitting **along certain specific directions (such as Γ-X)** with the inclusion of the SOC [see Fig. 4(b, e)].”.

In the revised Supplementary Materials, we have added Fig. R13 as new Fig. S15, and also included the related discussions.

Fig.3d,h: suggest to include exponential fits to the spatially decaying LDOS profiles. I understand that data quality is limited, but guides to the eye would still be helpful illustrate the energy-dependence of the imaginary wave vector for states decaying into the vacuum of the bulk (at least in Fig.3h).

Response: In the revised Fig. 3(d, h), we have included the exponential fits to the spatially decaying LDOS profiles for guiding to the eye, as shown in Fig. R14 below.

Fig. R14 Distance-dependent distribution of the edge state intensity from the step edge to bulk for phases I (a) and II (b). The top panel shows the line-scan profile extracted from the topographic images. The colored lines in the bottom panels are extracted from the dI/dV spectra at certain energies, with the fit to an exponential decay indicated by the black solid line for guiding eyes.

Once again, I believe that this local probe spectroscopy characterization of the QSH candidate ZrTe5 is well-worth publishing in Nature Communication, once all the above comments are successfully addressed.

Response: We thank the reviewer again for the very constructive comments and suggestions, which have been helpful in improving the manuscript with enhanced validity and better clarity. We hope our revised manuscript is now acceptable for publication in Nature Communications.

Reviewer #3 (Remarks to the Author):

This work is the first to achieve monolayer ZrTe₅, which is of some importance. However, quantum spin Hall insulators have been well studied in the last 20 years, and there is no new physics in this work. On the other hand, Bi on SiC substrate has already been found to be a quantum spin Hall insulator with a band gap of 800 meV, which is much larger than the band gap of monolayer-ZrTe₅. Therefore, the implementation of the present experiment has not exceeded the previous works. Therefore, in terms of the general interest, the impact and novelty, this work does not meet the high criteria of Nature Communications. Nevertheless, it might be suitable for specified journals. Here I raise two questions on the present manuscript even for its submission to other specified journals.

Response: We thank the reviewer for reviewing our manuscript. We would like to emphasize that, compared to the bismuthene/SiC work where a strong substrate strain (strong bonding) is present, the novelty and importance of the present work is that we manage to realize true vdW monolayers with weak substrate interactions, namely, quasi-freestanding monolayers, which can largely preserve their electronic and topological properties. Such true vdW monolayers are more promising than the strained ones for further explorations of emergent phenomena as well as QSH-based devices. To the best of our knowledge, the band gaps of the ZrTe₅ monolayers established in the present work hold the highest record in the experimentally realized quasi-freestanding QSH monolayers. Considering the rich phenomena discovered in bulk ZrTe₅, its monolayer counterparts are expected to host exotic physics as well, as to be explored in subsequent studies enabled by this demonstration.

(1) The calculated energy gaps of two different phases of monolayer ZrTe₅ are both smaller than the experimental value, which should be due to the underestimation of the energy gap by PBE type exchange correlation functional. The electronic structure of monolayer ZrTe₅ can be calculated by hybrid functional or MBJ method, which may give rise to the results being consistent with the experiment.

Response: We thank the reviewer for this suggestion. We have carried out additional first-principles calculations using the HSE06 hybrid functional with different mixing parameters (AEXX = 0.1, 0.2, 0.25, 0.3, and 0.5) to check the band gaps and topological properties of phase I and phase II. The largest band gaps are identified to be 160 and 157 meV for phase I and phase II, respectively, both with AEXX = 0.25 in the hybrid functional calculations. Figure R15 plots the band structures, corresponding Wilson loop, and edge states for both phases. For phase II, the band gap increases by 47 meV, while for phase I, the band gap remains nearly unchanged, which may be because its conduction band minimum shifts to a lower location close to the Γ point (marked in the red circle in Fig. R15(a)). Overall, the calculated band gaps remain smaller than the experimental values. Within the HSE06 functional calculations, we also confirm that

both phases persist as topologically nontrivial with $Z_2 = 1$ and exhibit topologically protected edge states (see Fig. R15(b,c,e,f)).

Figure R15 (a) Band structure of phase I within the HSE06 hybrid functional calculations, obtained from the Wannier interpolation. (b,c) The corresponding Wilson loop and edge states. (d-f) Same as (a-c) but for phase II.

Moreover, we have also performed the calculations with the MBJ potential to investigate the electronic properties of both phases, and the indirect band gaps are determined to be 21 and 0 meV for phase I and phase II, respectively, which are much smaller than the experimental values. This could be due to the fact that MBJ calculations are not self-consistent with respect to the total energy, making it challenging to characterize the actual band gaps of the topologically nontrivial ZrTe₅ monolayers.

On page 10 of the revised manuscript, we have added a new paragraph: “*In addition, we have carried out calculations using the HSE06 hybrid functional [55] to further examine the electronic and topological properties of both phases (see details in Supplementary Fig. S17). The band gap of phase I remains nearly unchanged at ~160 meV, while the band gap of phase II increases to ~157 meV. Both phases are verified to preserve their topologically nontrivial nature with $Z_2 = 1$ and exhibit topologically protected edge state.*”

In the revised Supplementary Materials, we have incorporated the related discussion, and also added Fig. R15 as new Fig. S17.

(2) The MBE-grown monolayer ZrTe₅ is different from the single-layer structure of

three-dimensional bulk materials. And two new phases were obtained in monolayer ZrTe₅, reported in the manuscript. Whether it is possible to obtain more new phases, resulting in quantum spin Hall insulators with larger energy gaps in monolayer ZrTe₅?

Response: We thank the reviewer for the suggestion. In our experiment, we have tried to grow ~100 samples on various substrates such as BLG/SiC, SrTiO₃, TiO₂, and silicon, etc., with finely tuned growth parameters. So far, ZrTe₅ monolayers were only successfully grown on the BLG/SiC substrate, and within the growth temperature window, phases I and II are the two dominant phases in the epitaxial ZrTe₅ monolayers. The bulk phase of ZrTe₅, is only occasionally observable in the second layer. However, the band gaps of the second-layer ZrTe₅ in phase I (~30 mV), phase II (~90 mV), and bulk phase (~80 mV), are all much smaller than the first-layer ZrTe₅ in phases I and II. As the growth temperature is increased to be higher than this temperature window, ZrTe₃ and ZrTe₂ phases start to emerge. Unfortunately, neither the epitaxial ZrTe₃ nor ZrTe₂ monolayer hosts a full band gap. On the other hand, we cannot rule out the appealing possibility that a 2DTI in other phase of ZrTe₅ may remain to be discovered under other growth conditions or on some other more desirable substrates.

In summary, we wish to sincerely thank all the three reviewers again, for their highly valuable contributions to the improvement of this paper. We hope the work is now ready to be published in Nature Communications.

List of changes made in the revised manuscript

1. Four new authors (X. Du, W. Li, L. Ji and F.-S. Li) and one new affiliation were added.
2. In the abstract, “240 meV” was changed to “254 meV”.
3. On page 3, 2nd paragraph, “exotic phenomena have been discovered in bulk ZrTe₅ [36-42]” was replaced by “exotic phenomena have been discovered in bulk ZrTe₅, such as chiral magnetic effect, anomalous Hall effect, 3D quantum Hall effect and log-periodic oscillations, etc. [36-42]”.
4. On page 3, 2nd paragraph, “exfoliation” was replaced by “synthesis”, and “freestanding” was replaced by “quasi-freestanding”.
5. On page 4, 1st paragraph, “As reference systems, the epitaxial ZrTe₂ monolayer has been investigated in previous studies [43,44]” were added.
6. On page 4, 3rd paragraph, “In order to successfully obtain the epitaxial of ZrTe₅ monolayer, fine tuning of the epitaxy parameters is found to be crucial. The substrate temperature for ZrTe₅ growth has to be set to slightly higher than that for Te crystallization on the surface to avoid the formation of redundant Te islands, and below those for ZrTe₃ and ZrTe₂ crystallization to avoid the formation of ZrTe₃ and ZrTe₂ monolayers” were added.
7. On page 5, 1st paragraph, “Our large-scale STM data further demonstrate that the orientations of ZrTe₅ monolayers in both phases are randomly distributed, regardless that of the BLG/SiC substrate, further indicating a rather weak interlayer vdW interaction (more data can be found in Supplementary Fig. S3). Therefore, these epitaxial ZrTe₅ monolayers are expected to host the quasi-freestanding electronic structures. It is noteworthy that the brighter contrast regions on top of the ZrTe₅ monolayers, as shown in Fig. 1(a), are the second layer ZrTe₅ islands. Besides the phases I and II that dominate the second layer ZrTe₅, the bulk phase of ZrTe₅ also starts to appear in the second layer (more experimental data and first-principles calculations for the bilayers can be found in Supplementary Figs. S4-S7). Due to the much smaller sizes and complicated interfacial structures, the second layer ZrTe₅ is out of the focus of the present work” were added.
8. On page 5, 2nd paragraph, “It is well known that the mismatch of lattice constants and rotational alignments between the epilayer and substrate, e.g., in the presence of a Moire superstructure, can give rise to a difference in the topographic appearances as well. However, the statistic of our STM data show that the morphologies of phases I and II are either independent on their lattice orientations. Moreover, both phases can even coexist in one monolayer island with the same orientation (more details can be found in Supplementary Fig. S3). Thus, it is concluded that the phases I and II of ZrTe₅ monolayers are intrinsically two distinct lattice structures, but not different appearances of the same lattice structure due to substrate interactions” was added.

9. On page 7, 3rd paragraph, “The typical dI/dV spectra, taken randomly on the phase-I terrace, but away from the step edge, are plotted together in Fig 3(a). A full gap of as large as ~220 mV is identified, ranging from ~-70 to ~+150 mV.” was changed to “The typical dI/dV spectra, taken randomly on the phase-I terrace, but away from the step edge, are plotted in Fig 3(a) (more data can be found in Supplementary Fig. S13). Taking the difference in energy between the intersection points for conduction and valence bands [48], a full gap of as large as ~235 mV is identified, ranging from ~-60 to ~+175 mV”.
10. On page 8, 1st paragraph, “Quantitatively, the determined bulk gap in phase II is ~ 240 mV, slightly larger than that in phase I” was changed to “Quantitatively, the determined bulk gap in phase II is ~ 254 mV, slightly larger than that in phase I”.
11. On page 8, 1st paragraph, “In addition, there seems to exist a resonance between -100 and -200 meV in the dI/dV spectrum taken at the terrace of phase II in Fig.3(e). It is noteworthy that the band gap of the epitaxial ZrTe₅ monolayer is much larger than that as measured on the cleaved surface of bulk ZrTe₅ [21,51]. Compared with the 3D weak topological nature of bulk ZrTe₅, the ZrTe₅ monolayer exhibits the true 2D nature of topological insulator in the monolayer limit.” was added.
12. On page 9, 2nd paragraph, “Moreover, since the calculated local density of states (LDOS) can make easy comparison with the STS data [54], we calculated the LDOSs of phase I and its topological edge states in semi-infinite nanoribbons perpendicular to the *a* axis. As seen in Fig. 4(d), the LDOS of the edge states exhibits a gapless ‘V’-shaped feature within the bulk band gap, which is well consistent with the experimental results shown in Fig. 3(b).” was added.
13. On page 10, 1st paragraph, “In addition, we found that the nearly flat Rashba bands along the *Y'*- Γ -*Y''* direction in the whole Brillouin zone contribute to a pronounced density of states near the energy window of [-200 meV, -100 meV], which may result in the experimentally observed resonance, as shown in Fig. 3(e) (details can be seen in Supplementary Fig. S15)” was added.
14. On page 10, 1st paragraph, “... *the symmetry of phase II prevents spin-orbit splitting with the inclusion of the SOC [see Fig. 4(b, e)].*” was replaced by “... *the symmetry of phase II prevents spin-orbit splitting along certain specific directions (such as Γ -*X*) with the inclusion of the SOC [see Fig. 4(b, e)].*”.
15. On page 10, 1st paragraph, “As shown in Figs. 4(h), the calculated LDOS of the topological edge states also exhibits a gapless ‘V’-shaped feature within the bulk band gap, well consistent with the experimental results shown in Fig. 3(f).” was added.
16. On page 10, a new paragraph, “In addition, we have carried out calculations using the HSE06 hybrid functional [55] to further examine the electronic and topological properties of both phases (see details in Supplementary Fig. S17). The band gap of phase I remains nearly unchanged at ~160 meV, while the band gap of phase II increases to ~157 meV. Both phases are verified to preserve their topologically

nontrivial nature with $Z_2 = 1$ and exhibit topologically protected edge state.” was added.

17. In the Methods, sample synthesis section, “The ratio of Zr:Te flux was carefully set to ~1:5 to 1:10, in order to provide sufficient reactive Te atoms and simultaneously avoid the abundant Te condensation on the surface.” was added.
18. In the Methods, a new section of XPS measurements was added.
19. New XPS results were added as Figs. 1(f) and (g).
20. Figures 3(a),(b),(e),(f) were replaced with new data, and Figs 3 (d),(h) were updated with exponential fitting results for guiding eyes. The corresponding captions were updated.
21. New Figs. 4(d),(h) were added and the corresponding captions were updated.
22. New references [43], [44], [48], [51], [54] and [55] were added.
23. In the Supplementary Materials, new Fig. S1, S2, S3, S4, S5, S6, S7, S15, S17 and table SII were added. The original Figs. 3(a),(e) in the main text were moved to supplementary Fig. S13.
24. All the other minor revisions were marked in red font in the main text.

Reviewers' Comments:

Reviewer #1:

Remarks to the Author:

The authors have addressed all the issues raised in my previous review and have modified the manuscript accordingly. The current version of the manuscript can be published in Nature Communications.

Reviewer #2:

Remarks to the Author:

All my comments have been addressed to my satisfaction. I recommend publication of this manuscript.

Reviewer #3:

Remarks to the Author:

I am satisfied with the authors' reply, and recommend the manuscript to the NC.